# Intermittent turbulence in the heliosheath and the magnetosheath plasmas based on Voyager and THEMIS data

Wiesław M. Macek[1,2], Anna Wawrzaszek[2], and Beata Kucharuk[1]

[1]Faculty of Mathematics and Natural Sciences, Cardinal Stefan Wyszyński University, Wóycickiego 1/3, 01-938 Warsaw, Poland
[2]Space Research Centre, Polish Academy of Science, Bartycka 18A, 00-716 Warsaw, Poland
*Correspondence to:* Wiesław M. Macek (macek@cbk.waw.pl)

**Abstract.** Turbulence is complex behavior that is ubiquitous in space, including the environments of the heliosphere and the magnetosphere. Our studies on solar wind turbulence including the heliosheath, and even at the heliospheric boundaries, also beyond the ecliptic plane, have shown that turbulence is intermittent in the entire heliosphere. As is known turbulence in space plasmas often exhibits substantial deviations from normal Gaussian distributions. Therefore, we analyze the fluctuations of plasma and magnetic field parameters also in the magnetosheath behind the Earth's bow shock. Based on THEMIS observations, we have already suggested that turbulence behind the quasi-perpendicular shock is more intermittent with larger kurtosis than that behind the quasi-parallel shocks. Following this study we would like to present detailed analysis of intermittent anisotropic turbulence in the magnetosheath depending on various characteristics of plasma behind the bow shock and now also near the magnetopause. In particular, for very high Alfvénic Mach numbers and high plasma beta we have clear non-Gaussian statistics in the directions perpendicular to the magnetic field. On the other hand, for directions parallel to this field the kurtosis is small and the plasma is close to equilibrium. However, the level of intermittency for the outgoing fluctuations seems to be similar as for the incoming fluctuations, which is consistent with approximate equipartition of energy between the oppositely propagating Alfvén waves. We hope that the difference in characteristic behavior of these fluctuations in various regions of space plasmas can help to detect some complex structures in space missions in the near future.

## 1 Introduction

Turbulence is complex behavior that is ubiquitous in space, including the solar wind, interplanetary, and interstellar media, as well as planetary and interstellar shocks (e.g., Bruno and Carbone, 2016). These shocks are usually collisionless and processes responsible for the plasma are substantially different from ordinary gases, see e.g., (Kivelson and Russell, 1995; Burgess and Scholer, 2015). Namely, the necessary coupling in plasma is usually provided by nonlinear structures at various scales, possibly exhibiting fractal or multifractal self-similarity properties (e.g., Burlaga, 1995; Macek, 2006). In addition, dissipation (so called quasi-viscosity) could often result from wave damping or other processes related to electric current structures. The mechanism of complexity of space and astrophysical plasmas is still a challenge to turbulence problems (Chang, 2015).

## 2 Multifractal Model

In our view, we should still rely on phenomenological models of intermittent turbulence, which can grasp multiplicative processes leading to complex behavior of the plasma in a simply way. As we have often argued (e.g., Macek, 2006, 2007; Macek and Wawrzaszek, 2009), the most useful concept for such a phenomenological study is a topological object namely the generalized two-scale weighted Cantor set — an example of multifractals — as described, for example, by Falconer (1990). The turbulence model based on this set is sketched here in Figure 1, as taken from Macek (2007). We see that at each step of constructing of the generalized Cantor set one needs to specify two scales $l_1$ and $l_2$ ($l_1 + l_2 \leq 1$) associated with probability measures $p$ and $1 - p$. In fact, fractals and multifractals could be considered as a convenient mathematical language useful for understanding dynamics of turbulence, as already postulated by Mandelbrot (1982). In fact, in this review we will provide some arguments that this surprisingly simple mathematical rule provides a very efficient tool for phenomenological analysis of complex turbulent media.

Moreover, for the two-scale weighted Cantor set model the singularity multifractal spectrum shown in Figure 2 can easily be calculated (e.g., Ott, 1993). In particular, the width of this universal function, $\Delta$, is obtained analytically by the following equation

$$\Delta \equiv \alpha_{\max} - \alpha_{\min} = D_{-\infty} - D_{\infty} = \left| \frac{\log(1-p)}{\log l_2} - \frac{\log(p)}{\log l_1} \right|. \tag{1}$$

Naturally, this quantity $\Delta$ is just the difference between the maximum and minimum dimensions related to the regions in the phase space with the least dense and most dense probability densities, and hence it has been proposed by Macek (2007) and Macek and Wawrzaszek (2009) as a degree of multifractality. Moreover, since this parameter $\Delta$ exhibits a deviation from a strict self-similarity it can also be used as a degree of intermittency, as explained in (Frisch, 1995, chapter 8). One can expect that in the solar wind $\Delta$ reveals various nonlinear phenomena, including nonlinear pressure pulses related to magnetosonic waves, as argued by Burlaga et al. (2003, 2007).

The other parameter $A$ describing the multifractal scaling, is the measure of asymmetry of the spectrum as defined by Macek and Wawrzaszek (2009)

$$A \equiv \frac{\alpha_0 - \alpha_{\min}}{\alpha_{\max} - \alpha_0}, \tag{2}$$

where $\alpha = \alpha_0$ is the point at which the spectrum has its maximum, $f(\alpha_0) = 1$. In particular, in a simpler case when $A = 1$ ($l_1 = l_2 = 0.5$) one-scale $p$-model is recovered (e.g., Meneveau and Sreenivasan, 1987), and for a monofractal the function in Figure 2 is reduced to a point.

In principle, for experimental time series one can recover the multifractal spectrum and fit to either well-known $p$ model or the more general two-scale weighted Cantor set model. For Voyager data this can be done in the following way. Namely, the generalized multifractal measures $p(l)$ depending on scale $l$ can be constructed using magnetic field strength fluctuations (Burlaga, 1995). Normalizing a time series of daily averages $B(t_i)$, where $i = 1, \ldots, N = 2^n$ for $j = 2^{n-k}$, $k = 0, 1, \ldots, n$

$$p(x_j, l) \equiv \frac{1}{N} \sum_{i=1+(j-1)\Delta t}^{j\Delta t} B(t_i) = p_j(l), \tag{3}$$

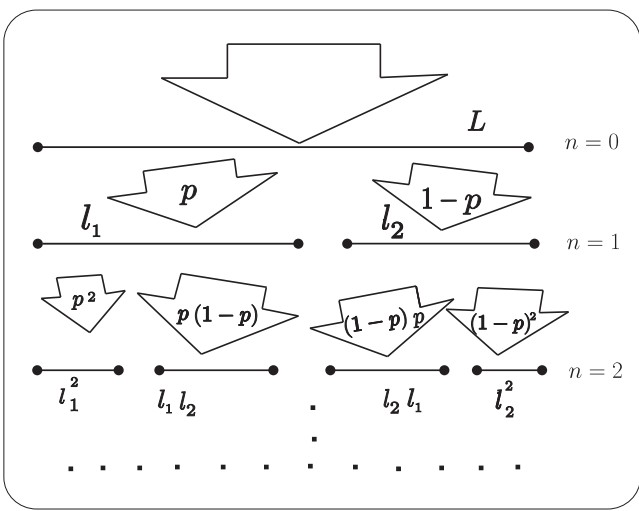

**Figure 1.** Two-scale weighted Cantor set model for asymmetric solar wind turbulence (Macek, 2007).

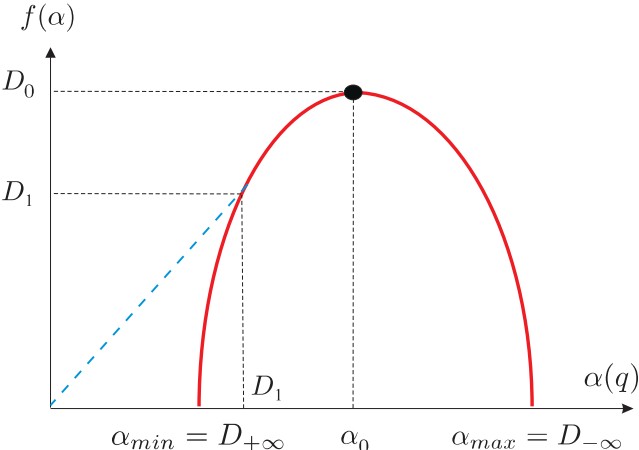

**Figure 2.** The singularity multifractal spectrum $f(\alpha)$ versus the singularity strength $\alpha$ with some general properties: (1) the maximum value of $f(\alpha)$ is $D_0$; (2) $f(D_1) = D_1$; and (3) the line joining the origin to the point on the $f(\alpha)$ curve, where $\alpha = D_1$ is tangent to the curve, as taken from (Ott, 1993).

is calculated with the successive average values $\langle B(t_i, \Delta t) \rangle$ of $B(t_i)$ between $t_i$ and $t_i + \Delta t$, for each $\Delta t = 2^k$ (Macek et al., 2011, 2012). When time series are obtained onboard spacecraft, it is usually possible to relate time dependence to space dependence by using Taylor (1938) hypothesis. Because the average solar wind speed $v_{sw}$ is much greater that the velocity of the space probe, we can argue that $p(x_j, l)$ can be regarded as a probability that at a position $x = v_{sw}t$, at time $t$, a given
magnetic flux is transferred to a spatial scale $l = v_{sw} \Delta t$.

In this way Burlaga (1995) has shown that in the inertial range the average value of the $q$th moment of $B$ at various scales $l$ scales as

$$\langle B^q(l) \rangle \sim l^{\gamma(q)}, \tag{4}$$

where the exponent $\gamma$ is related to the generalized dimension, $\gamma(q) = (q-1)(D_q - 1)$. If for a certain range of spatial scales $l$
corresponding to a given interval of time $\Delta t$ we have a straight line on a logarithmic scale using these slopes for each real $q$, the values of $D_q$ can be determined with Equation (4).

Alternatively, as explained by Macek and Wawrzaszek (2009), the multifractal function $f(\alpha)$ versus scaling index $\alpha$ shown in Figure 2, which exhibits universality of the multifractal scaling behavior, can be obtained using the Legendre transformation. It is worth noting, however, that we obtain this multifractal universal function directly from the slopes for a given scale range
using this direct method in various situations (see, Macek and Wawrzaszek, 2009; Macek et al., 2011, 2012, 2014).

## 3   Heliosheath Turbulence

The schematic of the heliospheric boundaries is shown in Figure 3. Voyager 1 entered the heliosheath after crossing the termination heliospheric shock at 94 AU in 2004, while Voyager 2 crossed this shock at 84 AU in 2007. It is generally accepted that after crossing the heliopause in 2012, the last boundary separating the heliosphere from the nearby interstellar medium,
the Voyager 1 has ultimately left the heliosphere, while the crossing of the heliopause by Voyager 2 is expected in the very near future.

### 3.1   Heliosheath Data

The main aim of our Voyager studies is to look at the measure of multifractal scaling in the heliosheath. Because in the distant heliosphere the magnetic fields have mainly azimuthal components one can use the magnitude of the magnetic fields $|\mathbf{B}|$ to
estimate the probability measures and using straight lines according to Equation (4) in a certain scale range, as those seen in Figures 1 and 2 of the paper by Macek et al. (2014), and in this way we can calculate the multifractal singularity spectrum. The results using the data gathered onboard both Voyager 1 and 2 spacecraft immersed in the heliosheath are presented in Figure 4, case (a) at 94–97 AU for the year 2005 and (c) at 105–107 AU for the year 2008 for Voyager 1, and Voyager 2 in case (b) at 85–88 AU for the year 2008 and (d) at 88–90 AU for the year 2009, respectively (Macek et al., 2012, Figure 5).
It seems that the two-scale weighted Cantor set model fits to the data better than the classical $p$-model. To support this result in a more quantitative way we have use the weighted $\chi^2$, consisting of a sum of squares of differences between the spectrum

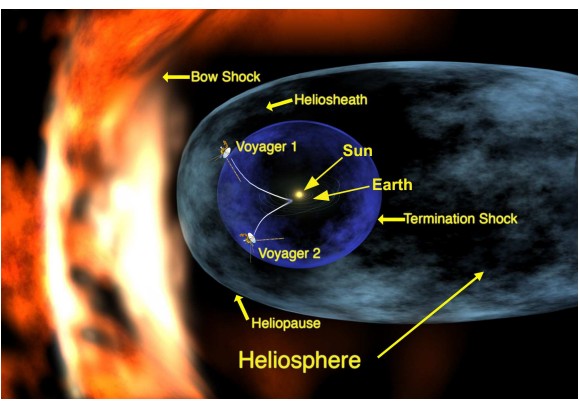

**Figure 3.** Schematic of the Heliospheric Boundaries [credit: NASA/Walt Feimer].

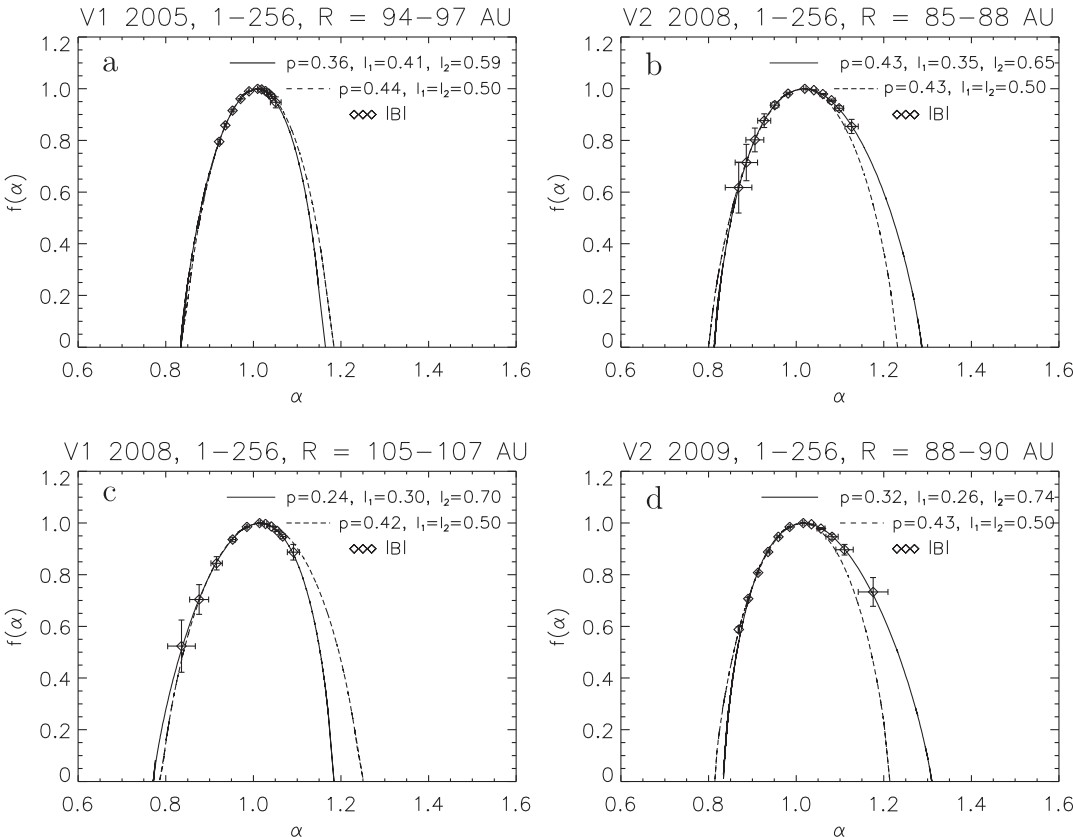

**Figure 4.** The singularity spectrum $f(\alpha)$ as a function of a singularity strength $\alpha$. The values are calculated for the weighted two-scale (continuous lines) model and the usual one-scale (dashed lines) $p$-model with the parameters fitted using the magnetic fields (diamonds) measured by Voyager 1 in the heliosheath at various heliocentric distances of (a) 94–97 AU and (c) 105–107 AU AU, and Voyager 2 at (b) 85–88 AU and (d) 88–90 AU, correspondingly, taken from (Macek et al., 2012).

**Table 1.** The measure of chi-square fitting for the weighted two-scale and the one-scale $p$-models.

| Fit quality | Voyager 1 | | Voyager 2 | |
|---|---|---|---|---|
| | 2005 | 2008 | 2008 | 2009 |
| P-model | 0.00240 | 0.00370 | 0.00190 | 0.03360 |
| Two-scale model | 0.00020 | 0.00036 | 0.00005 | 0.00069 |

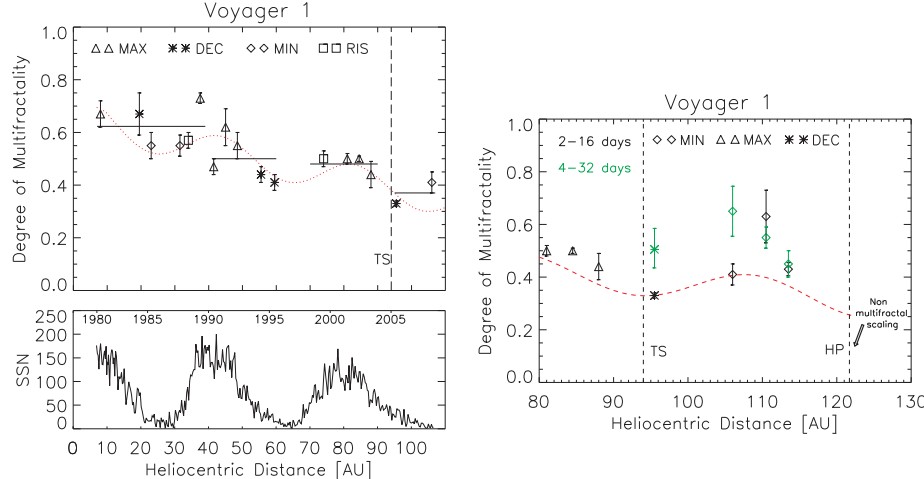

**Figure 5.** The parameter $\Delta$ quantifying multifractality in the heliosphere as a function of the distances from the Sun together with a periodic function shown by a continuous line during different phases of the solar cycle (SSN is the sunspot number) in the heliosphere (Macek et al., 2011) and in the heliosheath (Macek et al., 2014). The heliospheric termination shock (TS) and the heliopause (HP) crossings by Voyager 1 are indicated.

obtained from data and the model, each normalized to unit variance (e.g., Press et al., 1992). This measure of fit quality in the two cases is shown in Table 1. We see that the values obtained for the two-scale model are at least one order magnitude smaller than that for the standard $p$-model. This means that the generalized Cantor set model is in fact substantially better.

We have also calculated the degree of multifractality $\Delta$, as given in Equation (1), for Voyager 1 in the heliosheath depending on the heliospheric distances during different phases of the solar cycle: minimum (MIN), maximum (MAX), declining (DEC) and rising (RIS) phases, which is now demonstrated in Figure 5 (left panel). We clearly see that this multifractality measure obtained for the scale range of $\Delta t$ from 2 to 16 days decreases steadily with the heliospheric distance and is modulated by the solar activity following the sunspot numbers (SSN, indicated at the bottom panel), as taken from (Macek et al., 2011, Fig. 2). On the right panel we show $\Delta$ calculated in the heliosheath for two various scaling ranges from 2 to 16 and 4 to 32 days (cf. Macek et al., 2014). The crossings of the termination shock (TS) and the heliopause (HP) by Voyager 1 are indicated by vertical dashed lines. We see that in the heliosheath the degree of multifractality basically still follows the periodic dependence

| | $7-40\,\mathbf{AU}$ | $40-60\,\mathbf{AU}$ | $70-90\,\mathbf{AU}$ | $95-107\,\mathbf{AU}$ | $108-115\,\mathbf{AU}$ | $117-121\,\mathbf{AU}$ |
|---|---|---|---|---|---|---|
| $\Delta$ | $0.55-0.73$ | $0.41-0.62$ | $0.44-0.50$ | $0.33-0.42$ | $0.44-0.58$ | $0.22-0.35$ |
| $A$ | $0.47-1.39$ | $0.51-1.51$ | $0.47-0.96$ | $1.03-1.52$ | $0.88-0.98$ | $0.37-0.65$ |

**Figure 6.** The degree of multifractality $\Delta$ and asymmetry $A$ in the heliosphere as a function of the distances from the Sun. The termination shock (TS) and the heliopause (HP) crossings by Voyager 1 are also indicated (cf. Macek, 2012).

fitted inside the heliosphere (Macek et al., 2011, 2012). It is worth noting that after crossing the heliopause at $\sim$122 AU, the value of $\Delta$ suddenly drops to zero, and nonmultifractal (nonintermittent) smoothly varying magnetic fields are observed by Voyager 1, as indicated on the right panel of Figure 5, see (Macek et al., 2014). This would mean that the entire heliosphere with turbulent plasma inside is immersed in a relatively quiet ambient very local interstellar medium.

Naturally, the multifractal spectrum can be related to nonlinear Alfvén waves, associated with discontinuities or mirror mode structures due to some plasma instabilities or possibly current sheets (Borovsky, 2010; Tsurutani et al., 2011a, b) generated upstream of the termination shock, as discussed in our previous paper (Macek and Wawrzaszek, 2013). In this way we have applied the multifractal model (Macek, 2007; Macek and Szczepaniak, 2008) to solar wind turbulence in the entire heliosphere (Szczepaniak and Macek, 2008; Macek and Wawrzaszek, 2009; Macek et al., 2011, 2012), also beyond the ecliptic

plane (Wawrzaszek and Macek, 2010; Wawrzaszek et al., 2015), and even at the heliospheric boundaries (Burlaga et al., 2013; Macek et al., 2014), and have shown that turbulence could often be intermittent. By the way, it would be difficult to argue that there is an asymmetry in these spectra for the Voyager 1 data, but there are some deviations from symmetric spectrum for Voyager 2. In summary, the values of the degree of intermittency calculated ford our two-scale weighted Cantor set model are presented in Figure 6 (cf. Macek, 2012).

As is known turbulence in space and astrophysical plasmas exhibits deviations from normal distributions and these higher moments are often considered as signatures of intermittency. In particular, kurtosis – the fourth moment of the probability density function – is often used as a measure of intermittency (Bruno et al., 2003; Bruno and Carbone, 2013).

## 4 Magnetosheath Turbulence

Naturally, nonlinear structures responsible for turbulence have been already identified in planetary environments, in the solar

wind and also in the magnetosheath (e.g., Alexandrova, 2008). In particular, the magnetic fluctuations using Wind (Lion et al.,

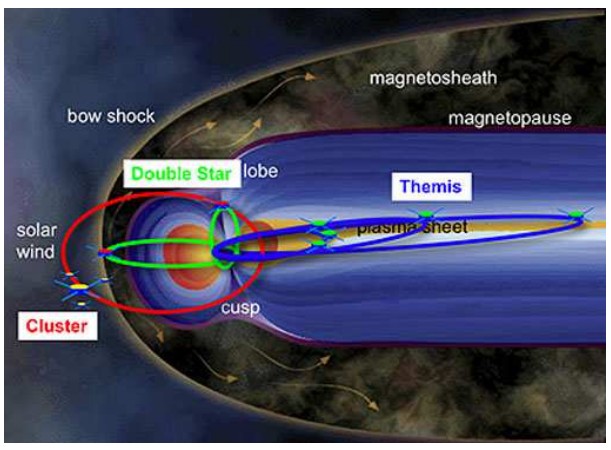

**Figure 7.** Schematic of THEMIS Mission [credit: NASA/ESA].

2016) and Cluster multi-spacecraft has been analyzed at ion scales (Yordanova et al., 2008; Roberts et al., 2016; Perrone et al., 2016, 2017). In addition, some results on very high resolution data on electron scales have recently been provided by Magneto-spheric Multiscale (MMS) Mission (Yordanova et al., 2016; Chasapis et al., 2017). Moreover, one basis of kinetic simulations by Karimabadi et al. (2014) one can suggest some interesting relationships of turbulent processes near shocks to reconnection processes. But in spite of progress in MHD simulations, including Hall effects, the physical mechanisms of turbulent behavior are still not sufficiently clear.

Various space missions provide unique observational data, which help to understand phenomena in our environment in space. In particular, the THEMIS mission was launched by NASA in 2007 in order to resolve macroscale phenomena occurring during substorms (Sibeck and Angelopoulos, 2008), as schematically presented in Figure 7. In addition, for the first time THEMIS data were used for analysis of turbulence at the terrestrial bow shock. Namely, we have suggested that turbulence behind the quasi-perpendicular shock is more intermittent with larger kurtosis than that behind the quasi-parallel shocks (Macek et al., 2015).

In this review paper besides turbulence in the heliosheath, as has already been discussed in Section 3, now in Section 4 we continue our study in the entire magnetosheath also near the magnetopause. However, since it would be difficult to obtain the full multifractal spectrum using the THEMIS data, at present we only examine how the degree of multifractality resulting in deviation from the normal distribution, which is also a level of intermittency, depends on the characteristics of the solar and magnetospheric plasmas (Macek et al., 2017). The data under study are briefly described in Subsection 4.1. In Subsection 4.2 we present the results of our analysis, showing in particular that at high Alfvénic Mach numbers turbulence becomes clearly intermittent. The importance of this intermittent behavior for space plasmas is underlined in Section 5.

## 4.1 Magnetosheath Data

We analyze various time samples acquired during the long period between 2008 and 2015 from THEMIS mission consisting of a quintet (A, B, C, D, and E) space probes (Sibeck and Angelopoulos, 2008), as listed in Table 2. We have selected the following 24 intervals in the magnetosheath (without any evident large-scale static plasma structures): 11 samples measured after crossing the bow shock, denoted by BS, and 13 samples obtained before leaving the magnetosheath, i.e. near magnetopause, denoted by MP. The time resolution here is 3 s and these samples taken in the Geocentric Solar Ecliptic (GSE) reference system are all (except no 11) longer than 4 h. Naturally, the length of each sample depends on the orbit of a particular probe immersed in the magnetosheath during some period of time. Please note that the time scales in the magnetosheath are much shorter than in the heliosheath.

Various characteristic plasma parameters, namely, the Alfvén Mach numbers, $M_A$, the plasma parameter beta, $\beta$, and the magnetosonic Mach number, $M_{ms}$, are calculated in the solar wind upstream: first before crossing the bow shock (before entering the magnetosheath) and next in the magnetosphere (before crossing the magnetopause). The plasma $\beta$ is the ratio of the thermal pressure $p$ to the magnetic pressure $\mathbf{B}^2/(2\mu_0\rho)$, where $\rho = mN$ is the mass density for ions of mass $m$ and the number density $N$ ($\mu_0$ denotes the permeability of free space).

All these 3 plasma parameters versus sample number are depicted in Figure 8. We see that the Alfvén Mach numbers can vary substantially with the limiting value of about 25 ($5 \leq M_A \leq 25$), and that in most cases cover $\beta$ is below 5 (only 3 cases are above 10). However the magnetosonic Mach numbers are rather moderate $3.6 \leq M_{ms} \leq 7.5$.

**Table 2.** List of selected interval samples (mm.dd.hh.MM).

| No. | THEMIS | Year | Location | Begin | End | $M_A$ | $\beta$ | $M_{ms}$ |
|---|---|---|---|---|---|---|---|---|
| 1 | THC | 2008 | BS | 06.27.18.30 | 06.27.23.15 | 14.13 | 2.98 | 7.50 |
| 2 | THC | 2008 | BS | 07.01.16.30 | 07.01.22.00 | 12.77 | 2.89 | 6.83 |
| 3 (a) | THC | 2008 | BS | 10.08.13.45 | 10.08.18.45 | 8.77 | 1.44 | 5.87 |
| 4 | THC | 2008 | BS | 10.18.12.45 | 10.18.17.30 | 9.97 | 2.85 | 5.33 |
| 5 (b) | THB | 2009 | BS | 06.10.16.45 | 06.11.02.00 | 11.66 | 3.51 | 5.84 |
| 6 | THC | 2009 | BS | 07.12.14.30 | 07.12.21.00 | 15.05 | 4.10 | 7.13 |
| 7 | THC | 2009 | BS | 07.31.08.30 | 07.31.13.30 | 8.43 | 1.79 | 5.30 |
| 8 | THC | 2009 | BS | 08.12.01.00 | 08.12.05.15 | 16.93 | 6.25 | 6.70 |
| 9 | THC | 2009 | BS | 08.19.17.00 | 08.19.22.00 | 6.50 | 1.18 | 4.57 |
| 10 | THB | 2010 | BS | 01.04.07.15 | 01.04.15.15 | 6.73 | 1.55 | 4.38 |
| 11 | THB | 2010 | BS | 04.13.09.30 | 04.13.13.15 | 7.15 | 1.05 | 5.20 |
| 12 | THC | 2008 | MP | 05.14.13.45 | 05.15.12.45 | 15.93 | 6.95 | 6.23 |
| 13 | THD | 2008 | MP | 09.13.15.15 | 09.13.22.00 | 22.00 | 12.45 | 6.53 |
| 14 | THA | 2009 | MP | 08.22.04.15 | 08.22.14.00 | 8.98 | 1.45 | 5.98 |
| 15 (c) | THA | 2009 | MP | 12.23.14.00 | 12.23.21.00 | 9.88 | 2.08 | 5.90 |
| 16 (d) | THA | 2010 | MP | 12.03.12.45 | 12.03.19.30 | 24.68 | 16.48 | 6.43 |
| 17 | THE | 2010 | MP | 12.03.13.00 | 12.03.20.30 | 21.10 | 11.97 | 6.35 |
| 18 | THA | 2011 | MP | 11.24.17.45 | 11.25.00.15 | 8.28 | 1.72 | 5.25 |
| 19 | THA | 2012 | MP | 01.15.16.30 | 01.16.00.15 | 11.00 | 2.97 | 5.83 |
| 20 | THD | 2012 | MP | 01.15.16.30 | 01.16.00.00 | 10.50 | 2.75 | 5.70 |
| 21 | THE | 2013 | MP | 01.31.17.45 | 02.01.00.45 | 11.63 | 3.62 | 5.75 |
| 22 | THD | 2014 | MP | 03.13.14.00 | 03.13.20.00 | 6.17 | 1.57 | 4.00 |
| 23 | THD | 2015 | MP | 03.23.00.45 | 03.23.05.30 | 5.27 | 0.51 | 4.40 |
| 24 | THE | 2015 | MP | 03.27.21.30 | 03.28.07.45 | 7.35 | 1.01 | 3.57 |

## 4.2 Results for the magnetosheath

Using the values of plasma and magnetic fields shown in Figures 1 and 2 of the paper by Macek et al. (2017) we can calculate the Elsässer variables, $\mathbf{z}^{\pm} = \mathbf{V} \pm \mathbf{V}_A$, where the characteristic Alfvénic velocity is given by $\mathbf{V}_A = \mathbf{B}/(\mu_0 \rho)^{1/2}$ (Elsasser, 1950). It worth noting that the sign is taken here relative to the local average magnetic field $\mathbf{B_o}$, which certainly depends on the time scale $\tau$ responsible for turbulence (Kiyani et al., 2013) as recently noted by Gerick et al. (2017). Because the time period during which this average background magnetic field is calculated, say $d\tau$, should be substantially larger than time scale of turbulence $\tau$, we have taken $d$ =10. By the way, in turbulence the dependence of statistical moments on spatial scales is often considered. For example, based on spacecraft measurements in the solar wind one can estimate spatial scales by using

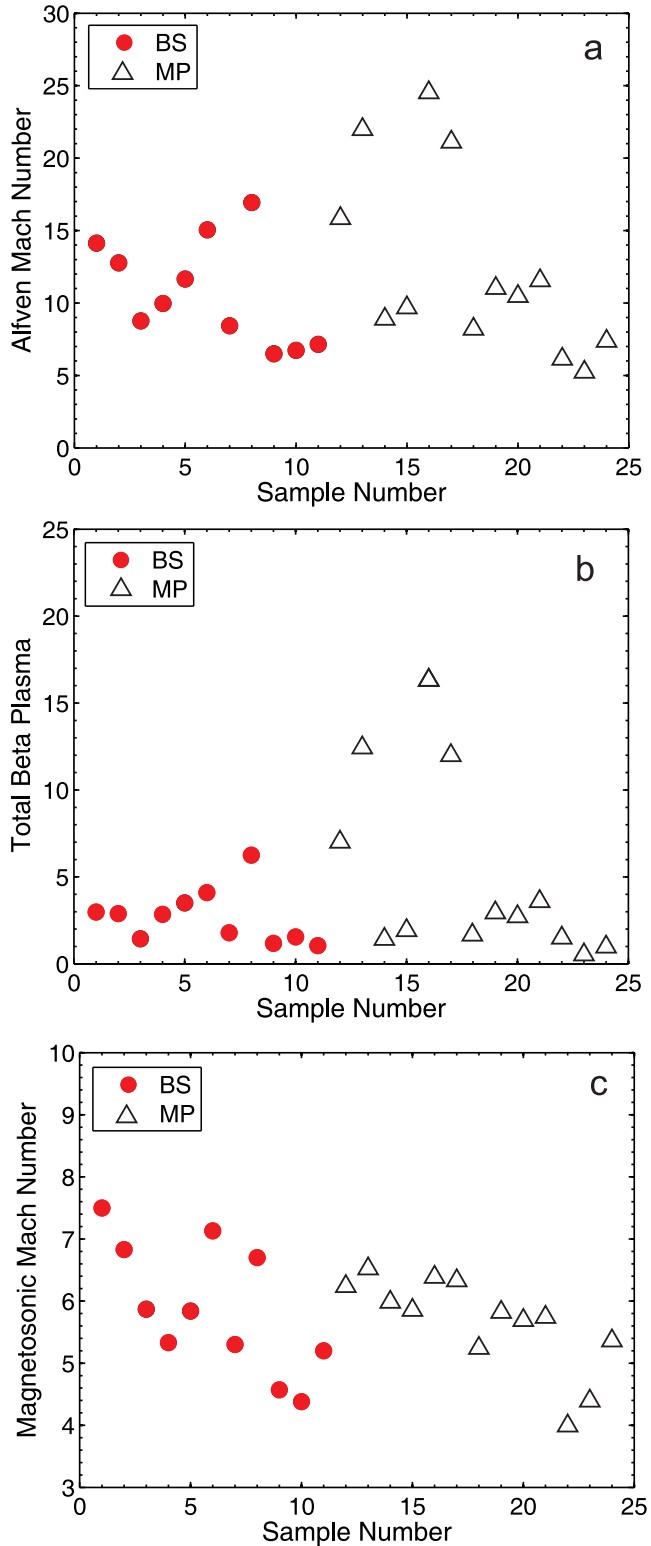

**Figure 8.** Alfvén Mach number (top), total plasma beta (middle) and magnetosonic Mach number (bottom) near the bow shock (BS, red circles) and the magnetopause (MP, white triangles).

the Taylor (1938) hypothesis (e.g., Macek and Wawrzaszek, 2009). However, in the magnetosheath the solar wind velocity is substantially reduced and this approach can be somewhat less certain (Mangeney et al., 2006), especially for some plasma parameters (Perri et al., 2017). Therefore, it is better to analyze directly time samples obtained onboard several space probes, as is in the case of THEMIS mission.

Now, following our previous work on THEMIS data, the kurtosis of the increments of the various components of both Elsässer vectors $\mathbf{z}^{\pm}$, $\delta\mathbf{z}^{\pm}(t,\tau) = \mathbf{z}^{\pm}(t+\tau) - \mathbf{z}^{\pm}(t)$, can be calculated for any given scale $\tau$, taken in units of time resolution (Macek et al., 2015, Equation (1)). As is known the Alfvénic increments perpendicular to the direction of $\mathbf{B_o}$ and the parallel compressive (slow-mode-like) increments should provide rather different contributions to the turbulent behavior of the solar wind plasma (e.g., Bruno et al., 2003; Oughton and Matthaeus, 2005). Therefore, we have performed our calculation in the

Mean Field (MF) coordinate system, as described by Bruno and Carbone (2013). Namely, the direction parallel to the local mean field $\mathbf{B_o}$ in the Geocentric Solar Ecliptic (GSE) system is taken along the versor $\hat{\mathbf{z}} = \mathbf{B_o}/B_o$ of the new MF reference system (the symbol ˆ is used for a unitary vector). This allows to calculate the parallel components of both Elsässer vectors $\delta z_{\parallel}^{+}$ and $\delta z_{\parallel}^{-}$. Next, in order to obtain two other components perpendicular to the field $\mathbf{B_o}$, $\delta z_{\perp 1}^{+}$, $\delta z_{\perp 1}^{-}$ and $\delta z_{\perp 2}^{+}$, $\delta z_{\perp 2}^{-}$, we take for the latter case the axis in the direction perpendicular to the plane containing the mean field $\mathbf{B_o}$ and the $\mathbf{X}$-axis in the GSE

system (taken here positive from the Sun), which is approximately consistent with the radial component of the mean solar wind velocity ($\mathbf{V}$), this means along $\hat{\mathbf{y}} = \hat{\mathbf{z}} \times \hat{\mathbf{x}}$. The remaining transverse components $\delta z_{\perp 1}^{\pm}$ are along $\hat{\mathbf{x}} = \hat{\mathbf{y}} \times \hat{\mathbf{z}}$ in this plane, which completes the right handed orthogonal MF system.

The obtained values of kurtosis of the increments of the fluctuations of the Elsässer variables for the outgoing and ingoing Alfvénic fluctuations, respectively $\mathbf{z}^{+}$ and $\mathbf{z}^{-}$, as observed by THEMIS in the magnetosheath near the bow shock (BS, red

circles) and the magnetopause (MP, white triangles), versus the Alfvén Mach number, the total plasma beta $\beta$, and the magnetosonic Mach number, corresponding to Figure 8, are presented in Figures 9, 10, and 11 for all other 24 cases listed in Table 2. The departure of the probability density functions from normal distributions for the selected four cases corresponding to Figures 1 and 2 of the paper by Macek et al. (2017), namely near the bow shock, cases (a) and (b), and near the magnetopause, cases (c) and (d), for a given time scale $\tau = 9$ s, are illustrated in Figures 12 and 13, respectively. The dependence of the kurtosis

on the time scale $\tau$ are depicted in the corresponding Figures 14 and 15.

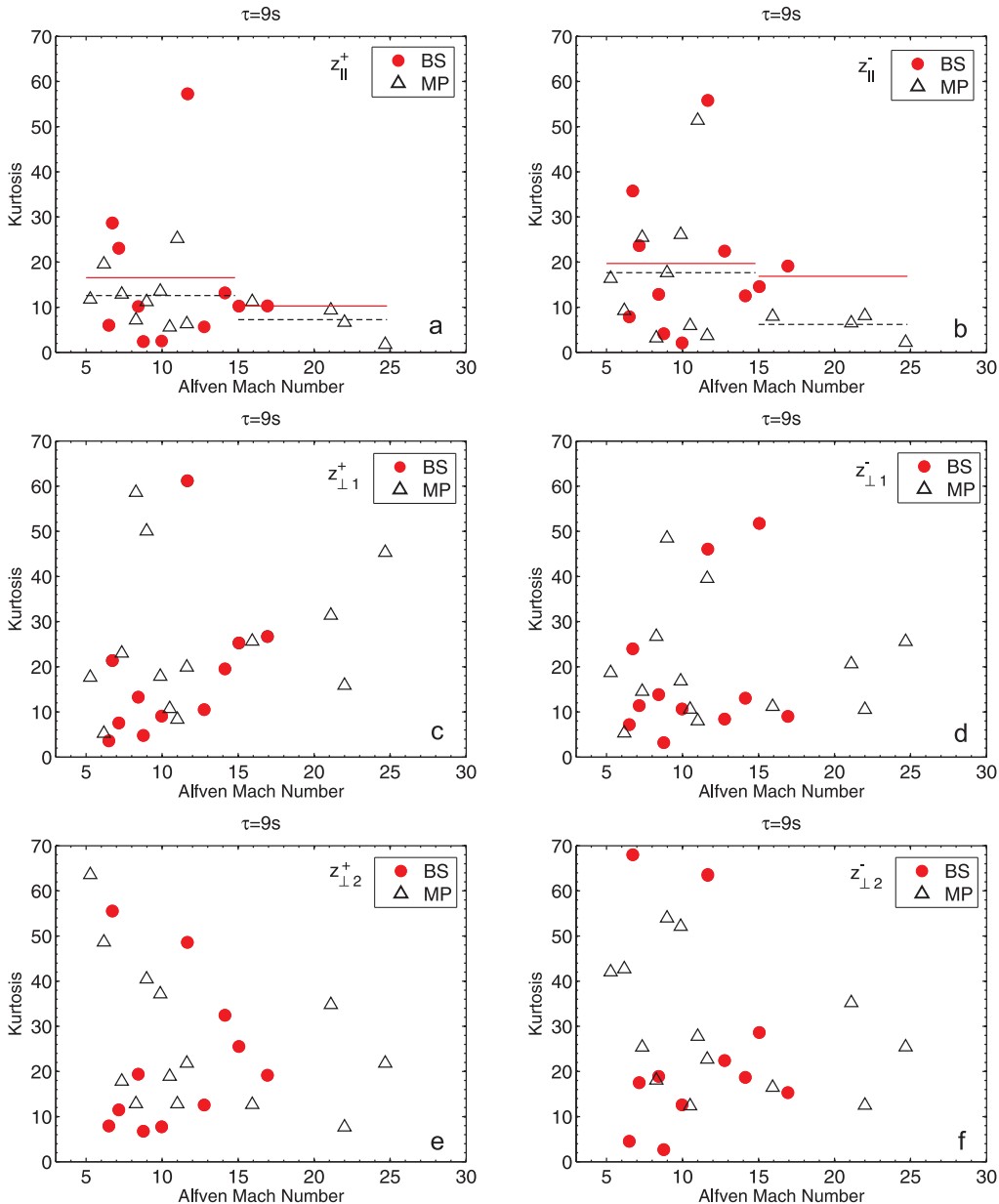

**Figure 9.** Kurtosis of the increments of the Elsässer vectors, $\delta\mathbf{z}^{\pm}$, in the magnetosheath versus the Alfvén Mach number for the components $\delta z^{\pm}_{\parallel}$ (parallel to the mean field vector $\mathbf{B_o}$, cases a and b) and two components perpendicular to the field: 1) $\delta z^{\pm}_{\perp 1}$ (in the plane containing $\mathbf{B_o}$ and $\mathbf{X}$-axis in GSE system, cases c and d) and 2) $\delta z^{\pm}_{\perp 2}$ (perpendicular to the plane, cases e and f) near the bow shock (BS, red circles), with averages marked by a continuous line, and near the magnetopause (MP, white triangles), marked by dashed lines, as observed by THEMIS for samples listed in Table 2.

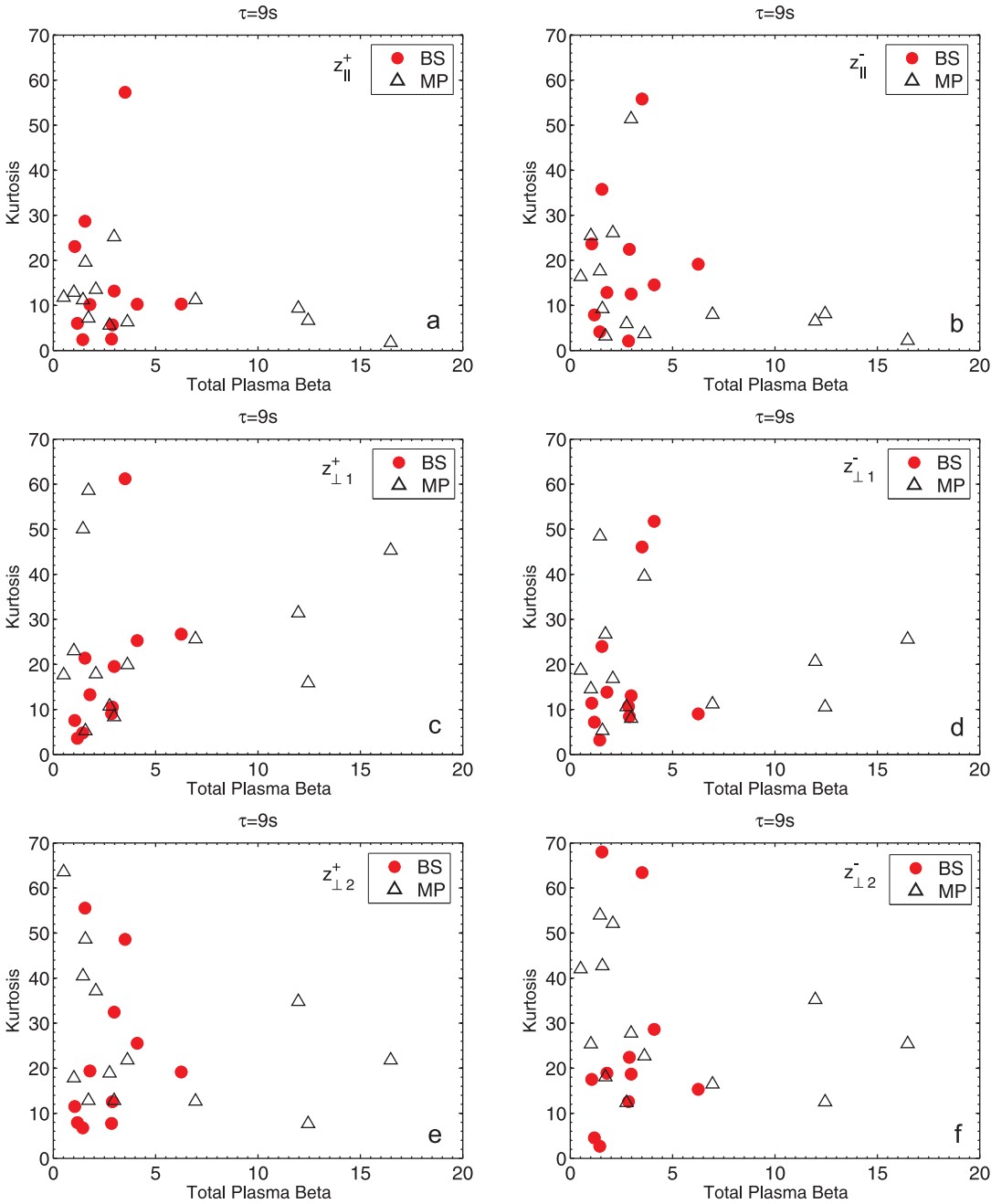

**Figure 10.** Kurtosis of the increments of the Elsässer vectors, $\delta \mathbf{z}^{\pm}$, in the magnetosheath versus the total plasma beta $\beta$ for the components $\delta z_{\parallel}^{\pm}$ (parallel to the mean field vector $\mathbf{B_o}$, cases a and b) and two components perpendicular to the field: 1) $\delta z_{\perp 1}^{\pm}$ (in the plane containing $\mathbf{B_o}$ and $\mathbf{X}$-axis in GSE system, cases c and d) and 2) $\delta z_{\perp 2}^{\pm}$ (perpendicular to the plane, cases e and f) near the bow shock (BS, red circles) and near the magnetopause (MP, white triangles), as observed by THEMIS for samples listed in Table 2.

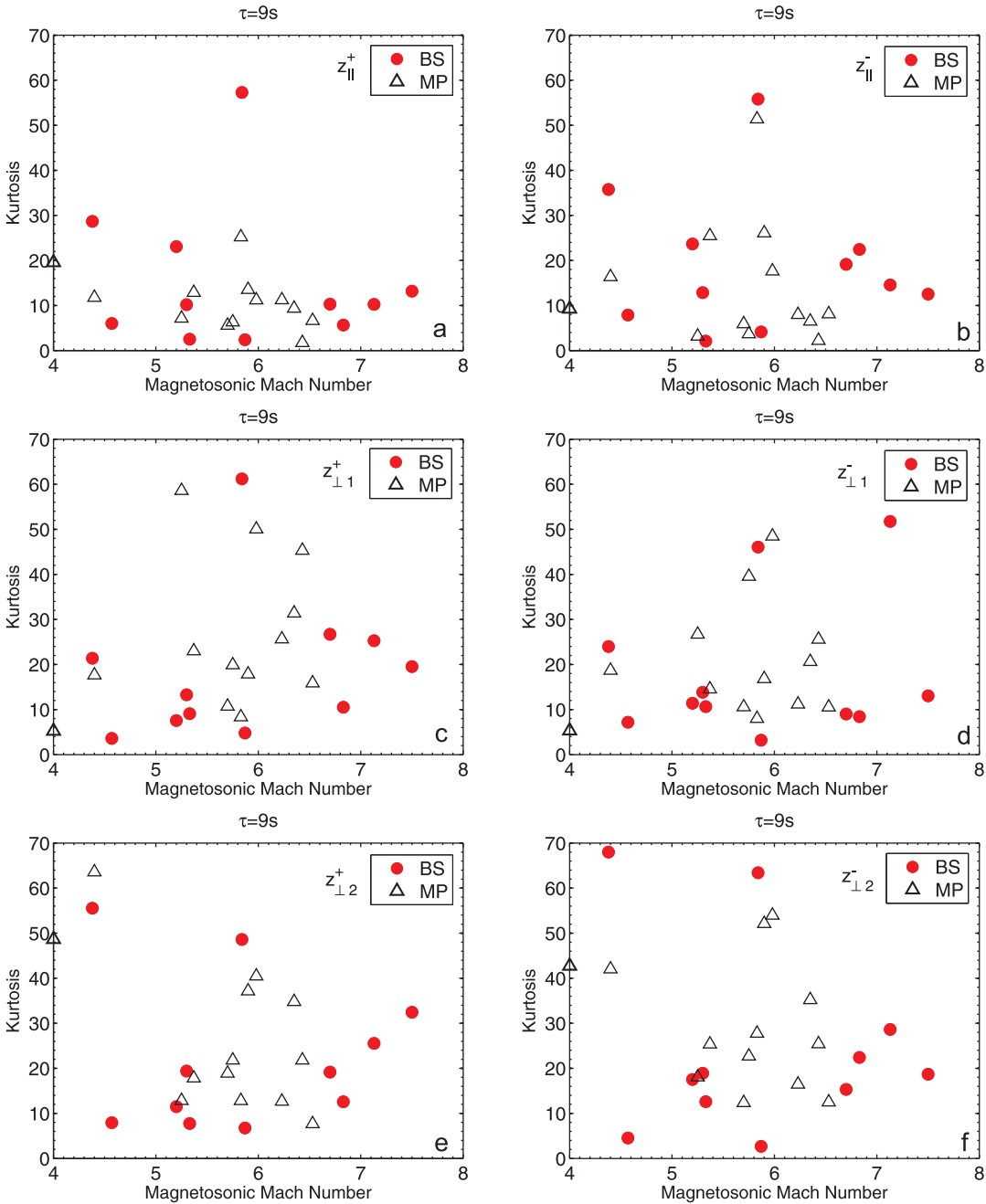

**Figure 11.** Kurtosis of the increments of the Elsässer vectors, $\delta\mathbf{z}^{\pm}$, in the magnetosheath versus the magnetosonic Mach number for the components $\delta z_{\parallel}^{\pm}$ (parallel to the mean field vector $\mathbf{B_o}$, cases a and b, and two components perpendicular to the field: 1) $\delta z_{\perp 1}^{\pm}$ (in the plane containing $\mathbf{B_o}$ and $\mathbf{X}$-axis in GSE system, cases c and d) and 2) $\delta z_{\perp 2}^{\pm}$ (perpendicular to the plane, cases e and f) near the bow shock (BS, red circles) and near the magnetopause (MP, white triangles), as observed by THEMIS for cases listed in Table 2.

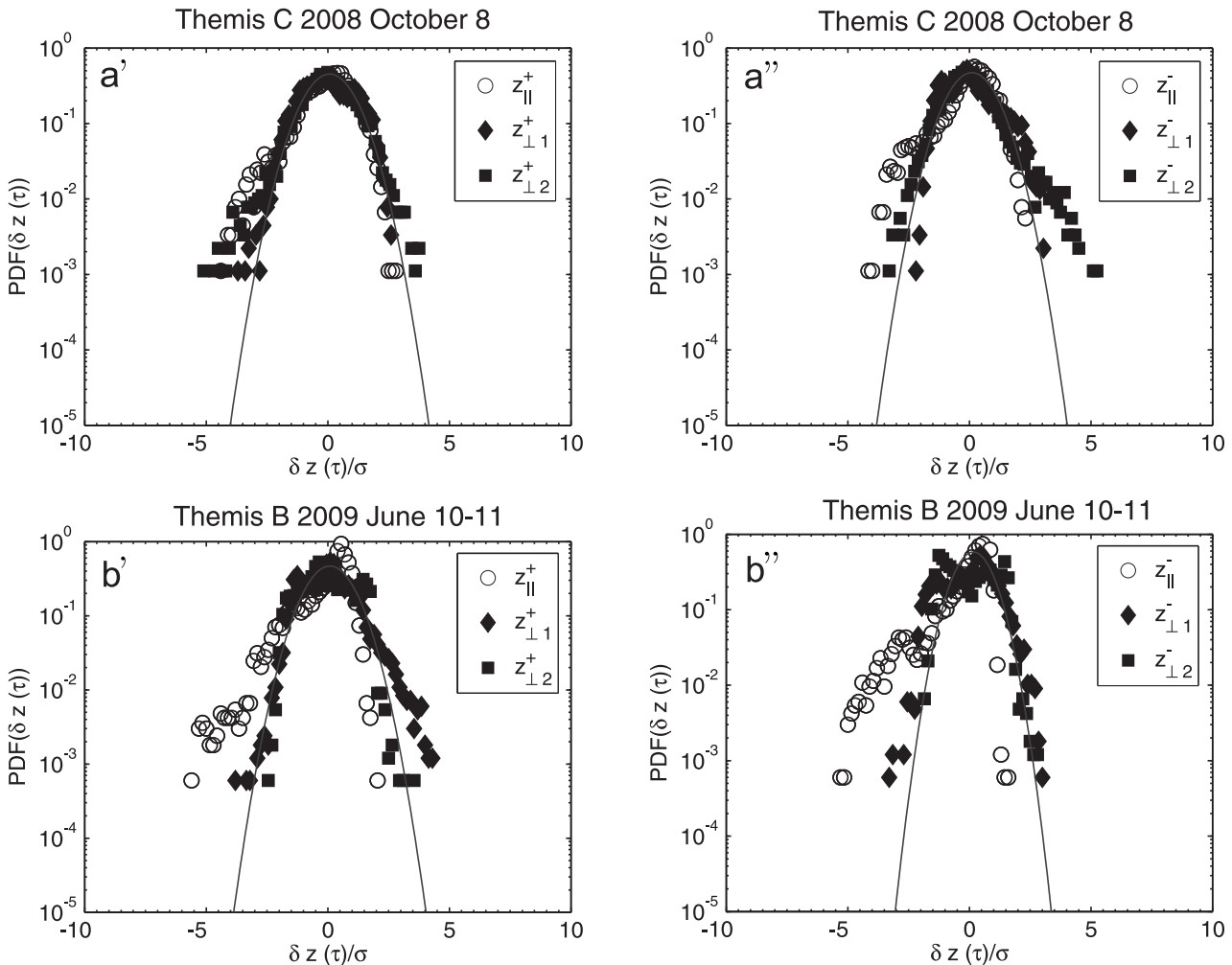

**Figure 12.** The probability density functions (PDF) of the increments of the parallel (white circles) and two perpendicular components (black diamonds and squares) of the Elsässer variables, $\delta\mathbf{z}^+$ (left panel) and $\delta\mathbf{z}^-$ (right panel), for a given time scale $\tau = 9$ s, near the bow shock, cases (a) and (b) in Table 2.

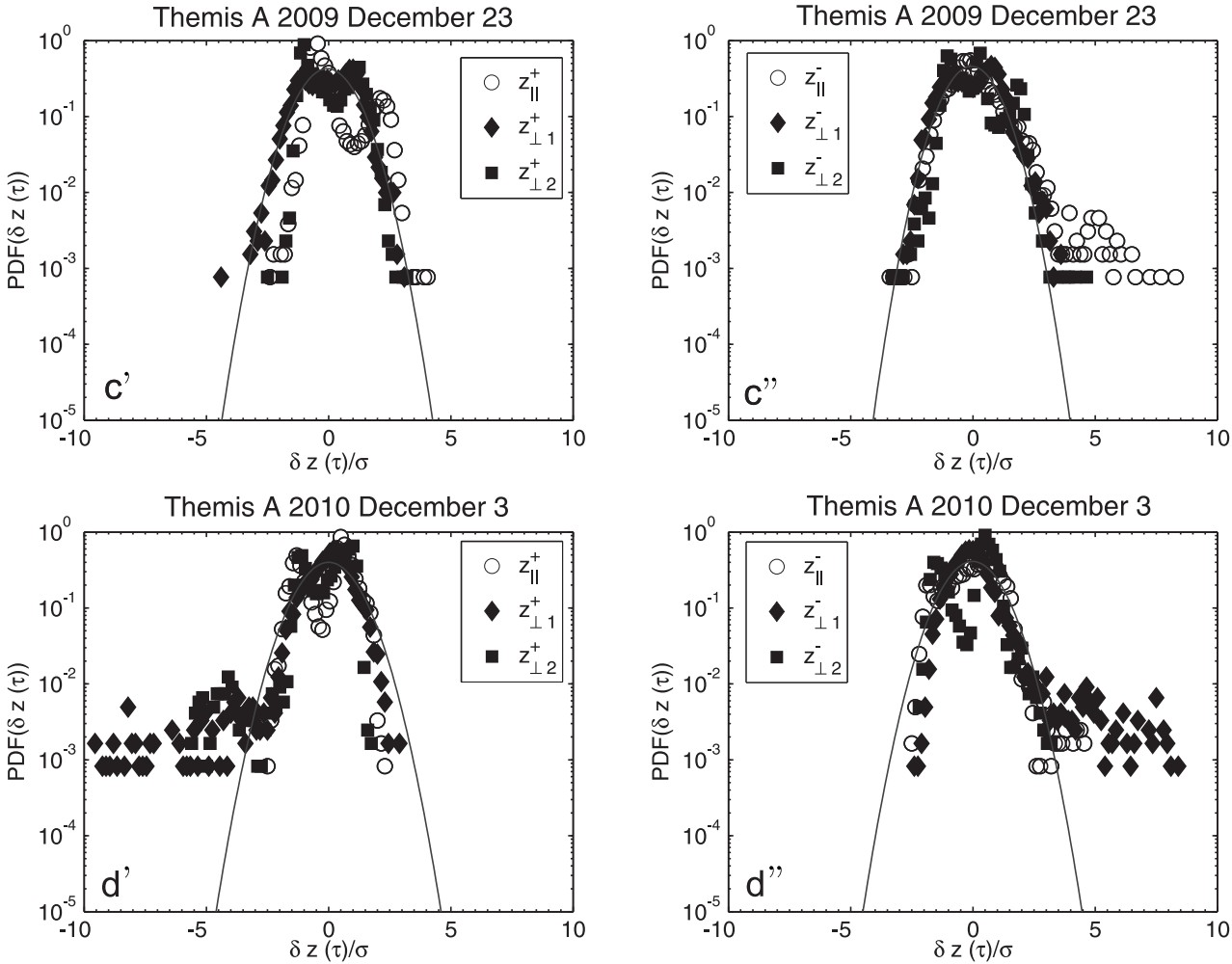

**Figure 13.** The probability density functions (PDF) the increments of the parallel (white circles) and perpendicular components (black diamonds and squares) of the Elsässer variables, $\delta\mathbf{z}^{+}$ (left panel) and $\delta\mathbf{z}^{-}$ (right panel), for a given time scale $\tau = 9$ s, near the magnetopause, cases (c) and (d) in Table 2.

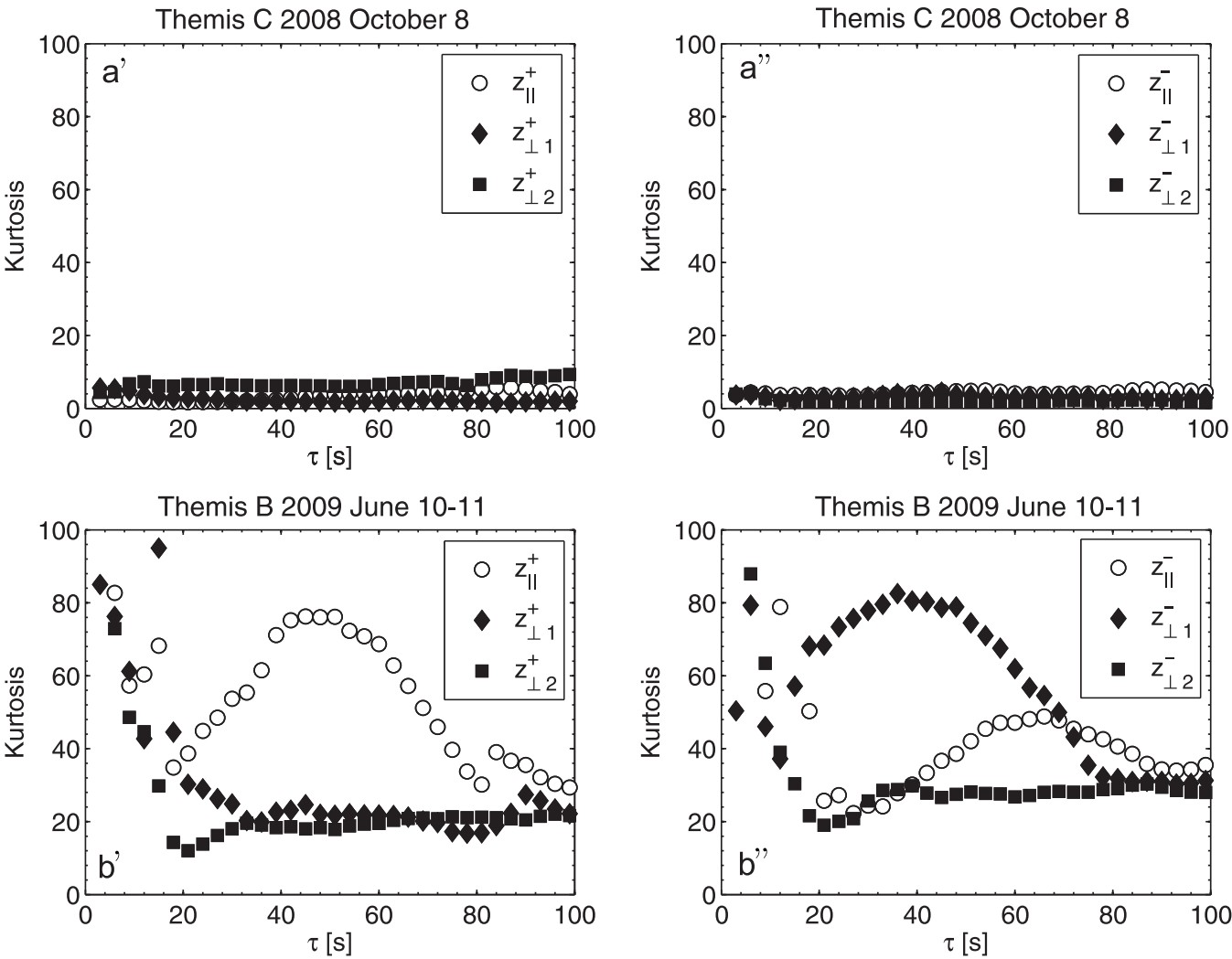

**Figure 14.** Kurtosis for the increments of the parallel (white circles) and two perpendicular components (black diamonds and squares) of the Elsässer variables, $\delta\mathbf{z}^+$ (left panel) and $\delta\mathbf{z}^-$ (right panel), as a function of time scale $\tau$ near the bow shock, cases (a) and (b) in Table 2.

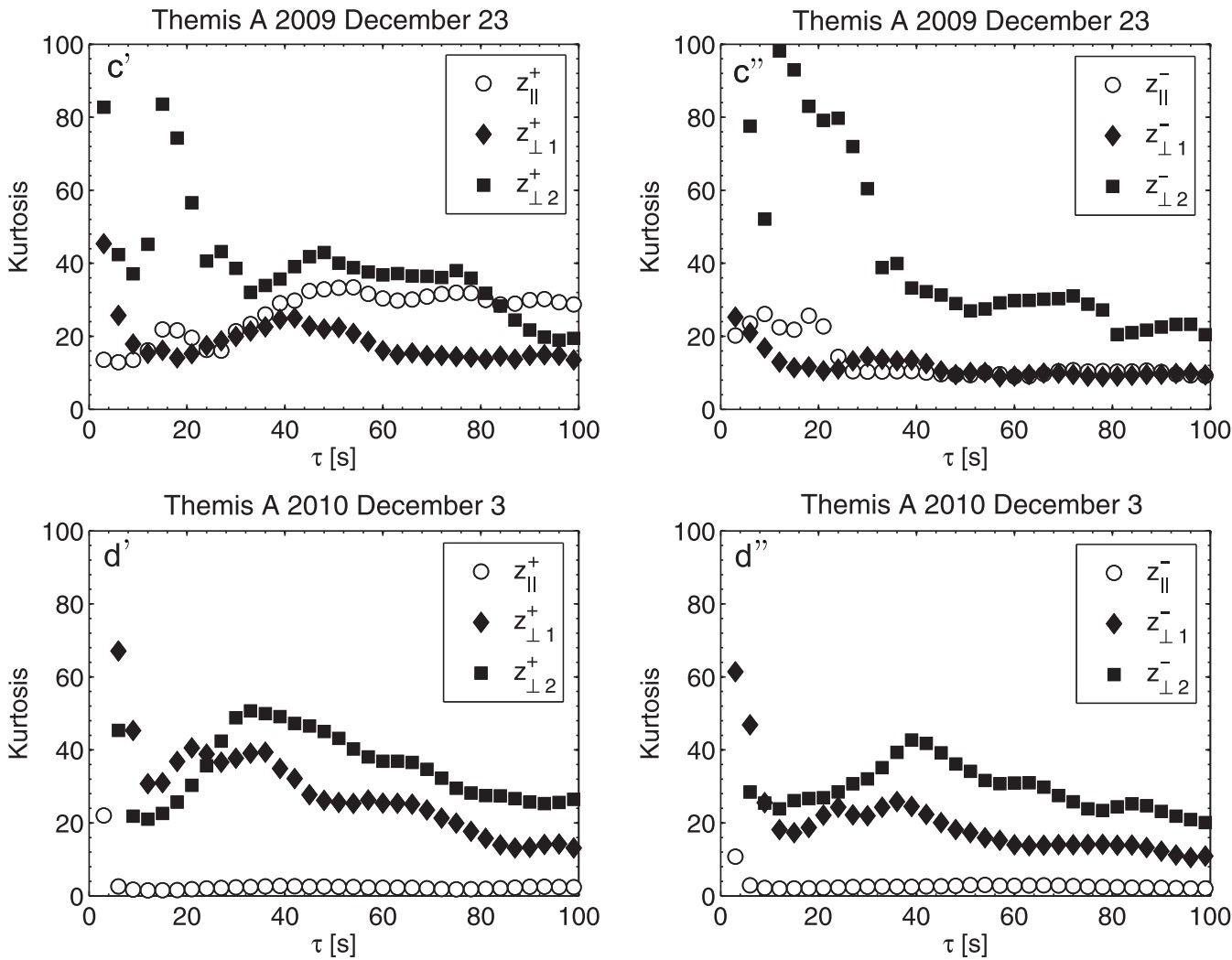

**Figure 15.** Kurtosis for the increments of the parallel (white circles) and perpendicular components (black diamonds and squares) of the Elsässer variables, $\delta\mathbf{z}^+$ (left panel) and $\delta\mathbf{z}^-$ (right panel), as a function of time scale $\tau$ near the magnetopause, cases (c) and (d) in Table 2.

Figures 9, 10, and 11 show kurtosis for the increments of all the components of the Elsässer vectors, for all the cases listed in Table 2, but for only one scale. Even though there is no very clear dependence on these plasma parameters one can notice that the value of kurtosis often decreases with Alfvénic Mach number along the local magnetic field and sometimes increases in the perpendicular directions. We see from Figure 9 that near the bow shock for the outgoing fluctuations kurtosis along the magnetic field $\delta z_\parallel^+$ somewhat decreases from $16.56 \pm 0.06$ at lower $M_A$ (bin: $5 \leq M_A \leq 15$) to $10.28 \pm 0.06$ at higher $M_A$ (bin $15 < M_A \leq 25$, even though we have only 2 points in this bin). But near the magnetopause, where we have more points in the letter bin, we can observe a clear significant decrease from $12.58 \pm 0.05$ to $7.25 \pm 0.05$. We have basically a similar behavior for the incoming fluctuations, $\delta z_\parallel^-$: a decrease from $19.69 \pm 0.06$ to $16.86 \pm 0.06$ at BS and a clear decrease from $17.66 \pm 0.05$ to $6.20 \pm 0.05$ at MP. Here the standard deviations of kurtosis are rather small of about 0.06, as calculated according to Press et al. (1992). However, for the transverse components $\delta z_{\perp 1,2}^\pm$ kurtosis seems to be more scattered and often rather increasing with the Alfvénic Mach number, exhibiting not only anisotropic but also seem to be non-gyrotropic with differences in two perpendicular components. We can see from Figure 10 that $\delta z_\parallel^\pm$ decrease with plasma $\beta$, approaching normal distribution for high $\beta$, when the thermal pressure dominates the plasma behavior. But we do not see any clear regularity for the dependence of $\delta z_{\perp 1,2}^\pm$ on $\beta$. It also seems from Figure 11 that the value of kurtosis is not very sensitive to the magnetosonic Mach number, but admittedly the range of this parameter considered in Table 2 are rather limited $3.6 \leq M_{ms} \leq 7.5$.

Additionally, for the four clearly quasi-perpendicular cases (illustrated in Figures 1 and 2 of the work by Macek et al. (2017), cases 3, 4, 15, and 16 listed in Table 2), the dependence of the parallel and perpendicular components of kurtosis on time scale $\tau$ for both the outgoing ($\mathbf{z}^+$) and incoming fluctuations ($\mathbf{z}^-$) is now presented in Figures 14 and 15, taken from (Macek et al., 2017). We can see that kurtosis behind the bow shock, Figure 14, cases (a) and (b) in Table 2, could sometimes (for $\beta \sim 1$) be smaller than that near the magnetopause, Figure 15, cases (c) and (d) in Table 2. We can generally notice only small differences between $\mathbf{z}^+$ and $\mathbf{z}^-$ and therefore the outgoing and ingoing fluctuations seem to be similar, what is roughly consistent with equipartition suggested by Tu et al. (1989). On the other hand, behind the bow shock for small plasma $\beta \sim 1$ (when the thermal pressure and the magnetic pressure are similar in the magnetized plasma), but with moderate Alfvénic Mach number $M_A \approx 9$, Figure 14 a' and a'', case (a) in Table 2, we see only small kurtosis with approximately Gaussian normal distribution (i.e., close to equilibrium). For similar $M_A \approx 12$ and somewhat higher plasma $\beta \sim 4$, Figure 14 b' and b'', case (b) in Table 2, both parallel and perpendicular components of the Elsässer vectors are active. A similar behavior is also observed near the magnetopause, Figure 15 c' and c'', case (c) in Table 2. Finally, it is worth noting that for the highest value of $M_A = 25$ and $\beta = 16.5$, as illustrated in Figure 15 d' and d'' (case (d) in Table 2), the perpendicular $\delta z_{\perp 1,2}^\pm$ components of fluctuations of Elsässer vectors are much larger than the parallel $\delta z_\parallel^\pm$ components. This exhibit a clear intermittent anisotropic turbulence with non-Gaussian probability distributions in transverse directions. On the other hand, the plasma along the local magnetic field is rather close to equilibrium.

Even though there is no clear regularity in Figures 14 and 15 showing dependence of the kurtosis on scale $\tau$, it seems that kurtosis near the bow shock (Figure 14) is rather similar to that near the magnetopause (Figure 15). Namely, based on Figures 9 and 10 it seems that near the bow shock (circles) the intermittency seems to decrease with the Alfvénic Mach number $M_A$ and decrease with the plasma beta $\beta$ near the magnetopause (triangles). We also see some difference between $\mathbf{z}^+$ and $\mathbf{z}^-$

in Figure 14 b' and b", behind the bow shock (BS), and some scatters near the magnetopause (MP) for small scales (Figure 15 c", for $\tau < 30$ s), and therefore could consider the other cases in Table 2. In fact, generally speaking, we have verified that the level of intermittency for the outgoing fluctuations $\mathbf{z}^+$ is usually similar as for the incoming fluctuations $\mathbf{z}^-$, which exhibits approximate equipartition of energy between these oppositely propagating Alfvén waves.

## 5 Conclusions

Using our weighted two-scale Cantor set model, which is a convenient tool to investigate the asymmetry of the multifractal spectrum, we confirm the characteristic shape of the universal multifractal singularity spectrum. In fact, as seen in Figure 4, $f(\alpha)$ is a downward concave function of scaling indices $\alpha$. We show that the degree of multifractality for magnetic field fluctuations of the solar wind falls steadily with the distance from the Sun and seems to be modulated by the solar activity also in the heliosheath. Moreover, we have considered the multifractal spectra of fluctuations of the interplanetary magnetic field strength before and after crossing of the heliospheric termination shock by Voyager 1 and 2 near 94 and 84 AU from the Sun, correspondingly.

Further, we have provided an important evidence that the large-scale magnetic field fluctuations reveal the multifractal structure not only in the outer heliosphere, but in the entire heliosheath, even near the heliopause. Naturally, the evolution of the multifractal distributions should be related to some physical (MHD) models, as suggested by Burlaga et al. (2003, 2007). The driver of the multifractality in the heliosheath could be the solar variability on scales from hours to days, fast and slow streams or shocks interactions, and other nonlinear structures discussed by Macek and Wawrzaszek (2013). In our view, any accurate physical model must reproduce the multifractal spectra. In particular, the observed nonmultifractal scaling after the heliopause crossing suggest a nonintermittent behavior in the nearby interstellar medium, consistent with the smoothly varying interstellar magnetic field reported by Burlaga and Ness (2014). We have identified the scaling region of fluctuations of the interplanetary magnetic field.

In fact, using our two-scale model based on the weighted Cantor set, we have examined the universal multifractal spectra before and after crossing by Voyager 1 the termination shock at 94 AU and before crossing the heliopause at distances of about 122 AU from the Sun. Moreover, inside the heliosphere we observe the asymmetric spectrum, which becomes more symmetric in the heliosheath. We confirm that multifractality of magnetic field fluctuations embedded the solar wind plasma for large scales decreases slowly with the heliospheric distance, demonstrating that this quantity is still modulated by the solar cycles further in the heliosheath, and even in the vicinity of the heliopause, possibly approaching a uniform nonintermittent behavior in the nearby interstellar medium. We propose this change of behavior as a signature of the expected crossing of the heliopause by Voyager 2 in the near future.

Regarding the magnetosheath we have shown that turbulence for small scales is intermittent in the entire magnetosheath, in regions near the bow shock and even near the magnetopause. In particular, we have found that near the magnetopause at very high Alfvénic Mach numbers $M_A$ and high plasma $\beta$ the probability density functions of compressive fluctuations parallel to the local average magnetic field should be nearly normal and close to equilibrium with small kurtosis, while in the transverse

Alfvénic turbulence resulting from nonlinear interactions is non-gyrotropic with large kurtosis for the Elsässer variables. These fluctuations are more intermittent than at the lower Alfvénic Mach numbers and plasma beta behind the bow shock. On the other hand, the level of intermittency for the outgoing fluctuations ($\mathbf{z}^+$) seems to be approximately similar as for the incoming fluctuations ($\mathbf{z}^-$). In view of the space investigation in the near future, including THOR mission (e.g., Vaivads et al., 2016), we expect that the difference in characteristic behavior of these fluctuations in various regions of the magnetosheath can help to identify some new complex structures in space plasmas.

*Acknowledgements.* We would like to thank the magnetic field instruments team of Voyager mission, the NASA National Space Science Data Center and the Space Science Data Facility for providing Voyager data. The research leading to these results has received funding from the THEMIS project during a visit of W.M.M. at the NASA Goddard Space Flight Center. We would like to thank the plasma and magnetic field instruments team of THEMIS mission for providing the data, which are available on-line from http://cdaweb.gsfc.nasa.gov. This work has been supported by the National Science Center, Poland (NCN), through grant 2014/15/B/ST9/04782.

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
