# Peer review of "Intermittent turbulence in the heliosheath and the magnetosheath plasmas based on Voyager and THEMIS data"

_Nonlinear Processes in Geophysics, 2017_

## Referee Comment (RC1) · Anonymous Referee #1 · 25 Sep 2017

Comments on "Intermittent turbulence in the heliosheath and the magnetosheath plasmas based on Voyager and THEMIS data" by Macek, Wawrzaszek and Kucharuk

The new results indicating that turbulence downstream of the quasiperpendicular portion of the bow shock is more intermittent with larger kurtosis than downstream of the quasiparallel portion of the bow shock. It is also found that for higher Mach number shocks, the intermittency for outgoing and incoming waves are similar, indicating equal energy densities in the two species. The work is well-formulated and straight-forward. I have only one suggestion. The authors should comment that the conclusions that they obtained from the Elsasser variable analysis depends on there not being any static

structures in the plasma medium.

Interactive
comment

---

## Referee Comment (RC2) · Anonymous Referee #2 · 2 Oct 2017

The manuscript is concerned with the analysis of intermittency of turbulent plasmas in two different environments, namely, the Heliosheath and the Earth's Magnetosheath. The authors have analysed data sets taken by various spacecrafts: Voyager 1 and 2 for the Heliosheath analysis, THEMIS for the Magnetosheath analysis. The main purpose of the paper is to evaluate the degree of intermittency. This is done using two different techniques. In the case of Heliosheath data, the predictions of a multifractal model (Two-scale weighted Cantor set) are fitted on Voyager data, thus deriving a measure (Delta) of intermittency. In the case of Magnetosheath the kurtosis of Elsasser variables is calculated finding that the level of intermittency is higher for higher Alfvenic Mach number. The paper is well written, the employed methods appear to be sound and

adequate, and the results are interesting. However, there are few points which the authors should consider before publication:

1) The analysis of Voyager data seems to be concerned with magnetic field magnitude data. This is not explicitly stated in the text, but it can be deduced from the legend of Fig. 4. Why do the authors use |B| data, instead of considering the behaviour of single magnetic field components? Is their analysis somehow related to the presence of compressive fluctuations (where |B| is modulated)? The authors should comment on that point.

2) From Fig. 4 it seems that the Two-scale weighted Cantor set model fits to the data better than the classical p-model. It would be interesting to support this result in a more quantitative way, by giving some measure of the fit quality in the two cases. Could the authors consider this suggestion?

3) In the analysis of THEMIS data, the authors have considered the magnitude of Elsasser variables, instead of single components. Which is the reason for this choice? Could the authors comment on this? (see point 1))

4) In turbulence the dependence of statistical moments (like kurtosis) on spatial scales is generally considered. When spacecraft measured are used, it is possible to "translate" time dependence into space dependence by using the Taylor hypothesis (like, for instance, in the solar wind). Could the author comment on how well the Taylor hypothesis is satisfied in the considered magnetosheath data set ?

---

## Referee Comment (RC3) · Anonymous Referee #3 · 7 Oct 2017

In this paper, the authors review their studies on the multifractal properties of the magnetic field in the heliosheath and magnetosheath. It is certainly an interesting paper, and worth publishing in this journal. However, some issues should be clarified before, most of them related to conclusions that are not so clearly supported by the results.

1. Please explain what does the "scaling ranges from 2 to 32 days" means. Although a reference is given, it would be better to understand the paper more self-consistently.

2. In the section on heliosheath data, it is said that the degree of multifractality basi-
cally "still follows the periodic dependence fitted inside the heliosphere". Since no mention has been given of a periodic fit inside the heliosphere before in the text, this sentence is not understood. What does the periodic dependence means?

Also, in the conclusions, the authors say that they have demonstrated that multifractality is modulated by the solar cycles further in the heliosheath. I guess this is the periodic funtion they refer to. If so, please state this more clearly at this point.

3. In the same sense, the caption of Fig. 5 refers to "a periodic function" which is not mentioned in the text.

4. Fig. 6 compares the degree of multifractality $\Delta$ and the asymmetry $A$ as obtained by the authors and Burlaga. Whereas $A$ follows a similar behavior in both references, having similar values, and increasing after the termination shock, $\Delta$ does not. Except at $95$–107 AU, Burlaga's values lie outside the range inferred from the two-scale model. Besides, $\Delta$ seems to be rather constant at the four distances from the Sun in the two-scale model, whereas Burlaga's values decrease in the heliosheath. Please comment on the differences.

5. Section on the magnetosheath, refering to Fig. 9. It is said that "the value of kurtosis often increases with Alfvénic Mach number". Please explain this observation, as it is not clear from Figs. 9 (a) and (b), in particular for the magnetopause (MP, triangles) points. In fact, a few lines later in the same paragraph, it is observed that the change is smaller, from 12.05 to 11.44. So it actually decreases.

A similar sentence is also in the conclusions, where it is said that "at higher Alfvénic Math numbers $M_A$ fluctuations are often somewhat more intermittent than at the lower numbers".

6. The discussion of Fig. 11 (last paragraph in Sec. 4) is not always clear. For instance, please clarify the sentence "but for cases (b) and (c) they are rather

similar". Are the authors refering to the kurtosis for $z^+$ and $z^-$? If so, this is not clear for Fig. 11 (c), for $\tau < 30$.

Also, later it is said that the level of intermittency for $z^+$ is usually similar to $z^-$. However, this is not seen in Fig. 11(c) for $\tau < 30$, and in Fig. 11(d) for most of the plotted range.

7. In the conclusions, the authors say that multifractality "falls steadily with the distance from the Sun". This does not seem to be supported by either Figs. 5 or 6.

8. Later, the authors state that they "have identified the scaling region of fluctuations". Please clarify the meaning of this sentence.

---

## Author Comment (AC1) · 13 Nov 2017

Thank you for a very nice report. The requested comment on there not being any static structure in the sample of the magnetosheath plasma is added on page 9 in line 6 of the revised manuscript, npg-2017-41_cor.pdf, where changes are marked in bold font.

Please also note the supplement to this comment:
https://www.nonlin-processes-geophys-discuss.net/npg-2017-41/npg-2017-41-AC1-supplement.pdf
[Figure]

**Supplement:**

[revised manuscript text omitted]

---

## Author Comment (AC2) · 13 Nov 2017

Thank you for your competent report with four comments, which have certainly been useful to improve the presentation of our results. These recommendations have been fully taken into consideration for a revision of the review. Please find the revised manuscript npg-2017-41_cor.pdf, with changes marked in bold typeface together with our response to points of your report.

Ad 1. The main aim of our Voyager studies is to look at the measure of multifractal scaling in the heliosheath. Till now only Voyager 1 has crossed the heliopause, the ultimate heliospheric boundary (see comment inserted on page 4 lines 18 – 21), but the

plasma data are not available on that spacecraft. Because in the distant heliosphere the magnetic fields have mainly azimuthal components one can use the magnitude of the magnetic fields to estimate the probability measure and using straight lines according to Equation (4) in a certain scale range similarly, as those seen in Figure 1 and 2 of the paper by Macek et al. (2014), and in this way we can calculate the multifractal spectrum, as shown in Figure 4 of the review. The proper explanations have also been inserted on page 4 lines 10 and 23ff of the revised manuscript. Naturally, as is now presented in Figure 5, the measure of multifractality is modulated by the solar activity as now clearly explained on page 6 lines 9 – 11.

Ad. 2. Thank you very much for this suggestion. We have added new Table 1 with chi-squares for the fits of the weighted two-scale Cantor set and the classical p-models. As we see these values are much lower for the two-scale model, which means that our model is substantially better than the p-model (see comments added on page 6, lines 1 – 6).

Ad. 3. Thank you very much for this important comment. Because the plasma in the magnetosheath is clearly anisotropic in the revised version we have analysed the fluctuations of the components of the Elsaesser variables in the plane perpendicular to the scale-dependent background magnetic fields and along the local average ambient magnetic fields. The obtained results are shown in Figures 9 – 15, as thoroughly discussed in subsection 4.2, where the modified text is marked in bold. The abstract and the conclusions are also consequently improved.

Ad. 4. Admittedly the Taylor hypothesis is used for solar wind data to relate the time scale to space scale needed to obtain the multifractal spectrum in the entire heliosphere according to Equation (4), see comment on page 4, lines 2 – 4. Because this approach is somewhat less certain in the magnetosheath (see our comment on page 10, lines 8 – 9 and page 12 lines 1 – 4 ), to obtain kurtosis by using THEMIS data in this region we analyze directly time samples.

Please also note the supplement to this comment:
https://www.nonlin-processes-geophys-discuss.net/npg-2017-41/npg-2017-41-AC2-
supplement.pdf

---

## Author Comment (AC3) · 13 Nov 2017

Thank you for your detailed report with 8 comments, which have certainly been useful to improve the presentation of our results in a more self-consistent way. We have taken into consideration all your points in the revised manuscript npg-2017-41_cor.pdf, where the changes are marked in bold typeface. Please also find the responses to each point of your report.

Ad. 1. In order to explain more clearly that within a given scaling range the pawer-law dependence of Equation (4) is satisfied several specifications are inserted on page 4 (lines 2 – 4, 9 – 10) and 6 (lines 10 – 11).

[Figure]

Ad. 2. As requested the periodic dependence of the degree of multifractality (parameter Delta) on the solar cycles are now added in Figure 5, taken from (Macek et al. 2011) and are thoroughly discussed on pages 6 and 7 what does the periodic dependence means.

Ad 3. The periodic function in question is now shown in Figure 5 (left panel). By the way, a sudden change of the this parameters to zero after crossing the heliopause is now also indicated on the right panel (which is a somewhat improved version of Figure 3 of Macek et. 2014 paper), as now discussed on page 7, in lines 4 − 8.

Ad. 4. Beside the weighted two scale Cantor model or the p-model one can take simple parabolic or cubic fits as used by Burlaga et al. (1993). Admittedly, this could result in somewhat different width of the multifractal spectrum, which is the parameter Delta in Figure 6. Because, contrary to the models discussed in this review, the polynomial fits have any clear theoretical interpretations, this values are not anymore given in Figure 6. By the way, this problem has already been discussed by Macek et al. (2014, first paragraph on page 2).

Ad. 5. Because the plasma in the magnetosheath is clearly anisotropic, following the suggestions of another referee, in the revised version we have analysed the fluctuations of the components of the Elsaesser variables in the plane perpendicular to the scale-dependent background magnetic fields and along the local average ambient magnetic fields. The obtained results are shown in Figures 9 − 11, as thoroughly discussed in subsection 4.2, where the modified text is marked in bold. The abstract and the conclusions are also consequently improved.

Ad. 6. Because of the anisotropic turbulence (see the reply to comment 5) the previous Figure 11 is replaced by Figure 14 and 15, as discussed in the revised manuscript on page 20.

Ad. 7. The main conclusion that multifractality falls steadily with the distance from the Sun is now clearly seen in Figure 5 (left panel), see reply to point 3.

Ad. 8. The scaling region of fluctuations is now discussed throughout the paper (pages 4 and 6), see reply to point 1.

Please also note the supplement to this comment:
https://www.nonlin-processes-geophys-discuss.net/npg-2017-41/npg-2017-41-AC3-supplement.pdf
* * *
[Figure]

**Supplement:**

[revised manuscript text omitted]